# Niacin Ameliorates Neuro-Inflammation in Parkinson’s Disease via GPR109A

**DOI:** 10.3390/ijms20184559

**Published:** 2019-09-14

**Authors:** Banabihari Giri, Kasey Belanger, Marissa Seamon, Eric Bradley, Sharad Purohit, Raymond Chong, John C. Morgan, Babak Baban, Chandramohan Wakade

**Affiliations:** 1Charlie Norwood VA Medical Center, Augusta, GA 30904, USA; bgiri@augusta.edu (B.G.); mseamon@augusta.edu (M.S.); spurohit@augusta.edu (S.P.); rchong@augusta.edu (R.C.); jmorgan@augusta.edu (J.C.M.); 2Department of Physical Therapy, Augusta University, Augusta, GA 30912, USA; 3Department of Physiology, Augusta University, Augusta, GA 30912, USA; kbelanger@augusta.edu; 4Department of Neuroscience, Augusta University, Augusta, GA 30912, USA; 5Edward Via College of Osteopathic Medicine, Greenville, SC 29303, USA; Ebradley@carolinas.vcom.edu; 6Department of Undergraduate Health Professionals, Augusta University, Augusta, GA 30912, USA; 7Department of Interdisciplinary Health Sciences, Augusta University, Augusta, GA 30912, USA; 8Department of Neurology, Augusta University, Augusta, GA 30912, USA; 9Department of Oral Biology and Diagnostic Sciences, Augusta University, Augusta, GA 30912, USA; bbaban@augusta.edu

**Keywords:** niacin, vitamin B3, GPR109A, macrophage, lipopolysaccharide, neuroinflammation, cytokine, neurodegenerative disease, Parkinson’s disease

## Abstract

In this study, we used macrophage RAW264.7 cells to elucidate the molecular mechanism underlying the anti-inflammatory actions of niacin. Anti-inflammatory actions of niacin and a possible role of its receptor GPR109A have been studied previously. However, the precise molecular mechanism of niacin’s action in reducing inflammation through GPR109A is unknown. Here we observed that niacin reduced the translocation of phosphorylated nuclear kappa B (p-NF-κB) induced by lipopolysaccharide (LPS) in the nucleus of RAW264.7 cells. The reduction in the nuclear translocation in turn decreased the expression of pro-inflammatory cytokines IL-1β, IL-6 in RAW264.7 cells. We observed a decrease in the nuclear translocation of p-NF-κB and the expression of inflammatory cytokines after knockdown of GPR109A in RAW264.7 cells. Our results suggest that these molecular actions of niacin are mediated via its receptor GPR109A (also known as HCAR2) by controlling the translocation of p-NF-κB to the nucleus. Overall, our findings suggest that niacin treatment may have potential in reducing inflammation by targeting GPR109A.

## 1. Introduction

Niacin or vitamin B3 has been shown to have a novel neuroprotective role in animal models of Parkinson’s disease (PD), stroke, traumatic brain injury, and multiple sclerosis [1,2,3]. Niacin has been studied clinically for over fifty years in the treatment of dyslipidemia [4], Crohn’s disease [5] and inflammatory bowel disease [6]. These studies have shown that niacin treatment improved vascular permeability, reduced apoptosis in epithelial cells, and most importantly suppressed the pro-inflammatory gene expression of macrophages (M1). Niacin triggers and boosts anti-inflammatory immune responses in humans and animal models of PD [7]. Specifically, niacin treatment had an anti-inflammatory polarization effect from pro-inflammatory macrophages (M1) to anti-inflammatory macrophages (M2) in PD subjects [8]. Neuroinflammation is one of the hallmarks of PD pathophysiology and although inflammation may be beneficial initially, prolonged and uncontrolled inflammation exacerbates brain damage [9]. Niacin may reduce neuroinflammation through the G-protein-coupled receptor, GPR109A, which has been noted to be up-regulated in PD patients [10].

GPR109A is ubiquitously expressed in a variety of cells including monocytes, leukocytes, neutrophils and macrophages [11]. This receptor is present in the brain and all other organs of humans and remains dormant until the onset of inflammation [12,13]. Interestingly, GPR109A was shown to be anti-inflammatory in colonic and retinal inflammation [14,15]. An up-regulation of GPR109A in the substantia nigra (SN) of PD patients has also been observed in postmortem PD subjects, making it an attractive target for niacin therapy [10].

In vivo and in vitro studies showed a robust activation of microglia that has been found in both, 1-methyl-4-phenyl-1, 2, 3, 6-tetrahydropyridine (MPTP)- and 6-hydroxydopamine (6-OHDA)-induced PD animal models [16,17]. Due to lineage proximity to microglia, macrophages have attracted increasing attention in relation to the onset and progression of PD. The interaction between microglia and macrophages, and their role in the progression of PD has gained recognition in the potential pathophysiology of PD [18]. In this study, we have utilized macrophages due to their higher expression of GPR109A [11] as compared to other similar available cell lines, such as human microglial cells. Moreover, macrophages are known to cross the leaky blood–brain barrier in PD to interact with microglia and stimulate the secretion of inflammatory cytokines to cause brain damage through neuroinflammation [9]. Uncontrolled release of inflammatory cytokines such as TNF-α, IL-1β, and IL-6 are key components contributing to neurodegenerative disease progression by inducing neuroinflammation [19,20,21].

Until now, it is unknown what specific role GPR109A plays in PD pathology or how niacin could possibly work to alleviate PD symptoms. In this study, we utilize lipopolysaccharide (LPS) as a mitogenic stimulant derived from Gram-negative bacteria [22,23], which induces the production of pro-inflammatory cytokines [24,25], as is similarly observed in PD subjects. Even though quality of life for PD subjects taking niacin supplements has been improved, the underlying molecular mechanism(s) has never been explored. Here, we focus on the molecular mechanism of niacin’s anti-inflammatory role in an in vitro study based on the observations shown in PD subjects [10]. Our aim was to elucidate the mechanism of niacin by examining its effects on macrophage cells treated with LPS. This study can provide useful information to understand the potential underlying mechanism of niacin on human PD subjects.

## 2. Results

### 2.1. Effects of Niacin on Parkinson’s Disease Subjects and an MPP+-Induced Parkinson’s Disease Model of Neuronal Cell Line N27

The increased expression of GPR109A was identified in human white blood cells of both Parkinson’s disease (PD) subjects compared to that of age-matched control subjects with polyclonal antibodies from Bioworld (Figure 1A). Figure 1A depicts the up-regulation of GPR109A for PD subjects compared with healthy, age-matched control subjects. In our ongoing clinical trial, the up-regulation of GPR109A in PD subjects was reduced in patients treated with niacin for at least one month (Figure 1A). Treatment of niacin with a daily dosage of 250 mg reduced the up-regulation of GPR109A in a time-dependent manner as shown in Figure 1A for half of a month and one month. The above results suggest that niacin treatment diminishes GPR109A up-regulation in PD subjects. To understand the fate of niacin in vivo, we have determined the levels of niacin and its metabolites in the plasma of the PD patients compared to age-matched control subjects (*n* = 24 each) as shown in Figure 1B. There is a significant decrease in the niacin metabolites of PD subjects compared to control subjects, indicating lower levels of niacin in PD patients compared to age-matched controls. Furthermore, Figure 1C, depicts the role of niacin to inhibit GPR109A expression, compared to the induced the expression of GPR109A by MPP+ alone shown in the upper panel. When treated with both MPP+ and niacin, the niacin was still able to down-regulate the GPR109A receptor. The lower panel shows the expression of β-actin as a loading control. The above results suggest that the up-regulation of GPR109A is mediated by MMP+ in vitro, mimicking findings observed in the PD subjects (Figure 1A). More importantly, niacin was able to decrease the up-regulation of GPR109A in two different scenarios (Figure 1A,C).

### 2.2. LPS Induced the Translocation of Phosphorylated Nuclear Factor-κB (p-NF-κB) to the Nucleus Inhibited by Niacin in Cultured RAW264.7 Cells

Figure 2A shows that LPS induced the translocation of the p-65-NF-κB subunit into the nucleus of RAW264.7 cells. Separation of nuclear and cytosolic fractions was determined by the nuclear marker protein H3 (Figure 2A) and cytosolic marker protein ρGDi-α (data not shown). Moreover, the nuclear translocation of p-NF-κB was compared between LPS and niacin treatment (Figure 2A). The results showed p-NF-κB translocation to the nuclei as shown in the presence of LPS. However, p-NF-κB translocation into the nuclear fraction is inhibited by niacin in a concentration-dependent manner. The rescue of p-NF-κB by niacin at two concentrations was compared quantitatively as represented in Figure 2B. The p-NF-κB was observed to be higher in cytosolic fractions (not shown) compared to the nuclear fractions in the niacin treated group which suggests that only a portion of p-NF-κB may be sufficient to induce inflammatory cytokine production [26,27]. To further confirm the nuclear translocation of p-NF-κB, we performed immunohistochemistry on LPS-induced cells in the presence and absence of niacin to determine the amount of p-NF-κB translocation which is shown in Figure 2C. Phospho-NF-κB was stained with p-NF-κB antibody followed by binding of the secondary antibody conjugated to Alexa 488. The distribution of p-NF-κB in the presence or absence of niacin in LPS-induced cells was observed (Figure 2C). The p-NF-κB remained ubiquitous throughout the cells in the absence of LPS (untreated and niacin conditions). LPS alone caused the accumulation of p-NF-κB in the nucleus of cells. However, niacin inhibited the nuclear translocation of p-NF-κB as shown in the Niacin + LPS panel. The nuclear translocation of p-NF-κB fluorescence intensity was determined by subtracting cytosolic fluorescence intensities from the normalized sum of nuclear and cytoplasmic fluorescence staining intensity. The intensity of p-NF-κB was higher (LPS panel) in the nuclei as determined by quantitation of the staining in Figure 2D. This experiment demonstrated that niacin inhibits the LPS-induced translocation of p-NF-κB to the nucleus.

### 2.3. Effect of Niacin on the Up-Regulation of IL-1β, IL-6, and TNF-α mRNA Induced by LPS

Unstimulated RAW264.7 shows no detectable levels of *IL-1β, IL-6*, and *TNF*-α transcripts in RAW264.7 cells in the absence of LPS (Figure 3A). After stimulation with LPS, increased levels of IL-1β, IL-6, and TNF-α transcripts were observed (Figure 3A). Pre-treatment with niacin significantly reduced the expression of *IL-1β* and *IL-6* transcripts in the presence of LPS. We did not see any differences in the expression of *TNF*-α transcript after LPS stimulation in response to niacin. Figure 3B shows the quantitation of cytokine transcripts shown in Figure 3A, and its fold changes are compared between each cytokine stimulated by LPS in the presence and absence of niacin. The control and niacin alone are not compared because they indicate the basal level of expression.

The above results, at least in the context of IL-1β and IL-6, suggest that the nuclear translocation of p-NF-κB induces pro-inflammatory cytokine transcript expression which is attenuated by niacin pre-treatment. Next, we had to determine the involvement of GPR109A in niacin preventing p-NF-κB nuclear translocation and subsequent inflammatory cytokine up-regulation.

### 2.4. Silencing of GPR109A by Gene-Specific siRNA in RAW264.7 Cells

The expression of GPR109A protein after knockdown by gene-specific silencing by siRNA is shown in Figure 4A. Western blot analysis was performed 48 h after the transfection to evaluate siRNA efficiency of knockdown of GPR109A protein expression. Figure 4A shows that the level of GPR109A knockdown was significantly decreased by 70% through GPR109A siRNA compared to scrambled siRNA. Densitometry analysis of GPR109A expression relative to the internal control β-actin is shown in Figure 4B. The GPR109A expression in the cells treated with scrambled siRNA was not affected compared to that of untreated cells.

### 2.5. Niacin Inhibits LPS-Induced Shuttling of p-NF-κB via GPR109A

Figure 5A demonstrates the translocation of p-NF-κB to the nucleus. The p-NF-κB was accumulated after LPS stimulation. Niacin significantly reduced the nuclear translocation of p-NF-κB in scrambled siRNA in the presence of LPS as shown in lane 4 compared to lane 3. In GPR109A siRNA knockdown cells, the LPS-induced nuclear translocation of p-NF-κB was reduced (lane 7 compared to lane 3), however, the nuclear translocation of p-NF-κB was not further reduced by niacin treatment shown by lane 8 compared to lane 7 under GPR109A knock down condition. Therefore, niacin failed to inhibit the nuclear translocation of p-NF-κB when GPR109A levels were substantially decreased. Additionally, the nuclear translocation of p-NF-kB was quantitated by densitometry scanning of band intensity which is depicted in Figure 5B. There was a significant fold change in the nuclear translocation of p-NF-κB between the scrambled and GPR109A siRNA knockdown in the presence of LPS. A significant change difference was also observed between scrambled and GPR109A siRNA knockdown in the presence of LPS with niacin. However, there was no noticeable difference of p-NF-κB nuclear translocation observed with LPS-treated cells between the presence and absence of niacin under GPR109A knockdown conditions. This result suggests that niacin–GPR109A coupling, at least in part, plays an important role in the inhibition of shuttling inflammatory transcription factor p-NF-κB from the cytosol to the nucleus.

### 2.6. Niacin Failed to Inhibit LPS-Induced Inflammation Response in GPR109A Knockdown Cells as Determined by Inflammatory Cytokines

Figure 6 shows the pre-treatment with niacin on the LPS-induced production of TNF-α, IL-1β and IL-6 in both scrambled siRNA and GPR109A siRNA-transfected RAW 264.7 cells. The expression of pro-inflammatory cytokines IL-1β, IL-6, and TNF-α was not decreased in the GPR109A knockdown cells in response to LPS compared to scrambled siRNA control (lane 7, 8 vs. Lane 3, 4). Moreover, pre-treatment of niacin inhibited LPS-induced pro-inflammatory cytokine production for IL-6 and IL-1β in scrambled siRNA, but not in GPR109A knockdown cells. Niacin failed to inhibit the LPS-induced production of pro-inflammatory cytokines IL-1β, and IL-6 at a reduced level or in the absence of GPR109A receptors. This establishes the role of GPR109A-mediated inflammation; therefore, niacin requires GPR109A for its anti-inflammatory action. This suggests the involvement of GPR109A in the regulation of inflammatory cytokines upon LPS stimulation, mediated through p-NF-κB nuclear translocation. Since TNF-α and IL-10 (anti-inflammatory) expression levels remained unchanged between LPS with the presence and absence of niacin treatments, this suggests that TNF-α (pro-inflammatory) may not be involved in the Niacin-GPR109A-NF-κB pathway. Our results support that niacin not only reduces GPR109A up-regulation, but also reduces pro-inflammatory cytokines by inhibiting p-NF-κB translocation to the nucleus.

## 3. Discussion

Recently, niacin has been investigated in a clinical trial with PD subjects [28], resulting in improved quality of life with daily supplementation. This investigation is important because PD is the second most prevalent neurodegenerative disease worldwide, next to Alzheimer’s disease (AD), affecting 1% of the world population [3]. Although enormous progress has been made in the treatment of PD over the past half century, even the gold standard drug levodopa leaves many non-motor symptoms of PD untouched [29]. Here our results suggest that niacin may reduce PD symptoms and have beneficial effects on subjects through interaction with anti-inflammatory receptor, GPR109A.

We have observed up-regulation of GPR109A in white blood cells (WBCs) [30] of PD patients, which responds to low dose niacin intervention by diminishing its expression [10]. Now in this study, along with up-regulation of GPR109A in WBCs, reduced levels of niacin metabolites in plasma were also noted in PD patients. Niacin was able to reduce GPR109A expression at half a month and one month supplementation. Similarly, we showed that GPR109A up-regulation is observed after MPP+ treatment in a dopaminergic rat neuronal cell (N27), another in vitro PD model (Figure 1C). MPP+ has been studied in mouse models of PD, showing initiation of an inflammatory cascade characterized by a series of inflammatory cytokine production in the SN of mouse brains [31]. Up-regulation of GPR109A by MPP+ in vitro was inhibited by niacin similarly to the LPS-induced up-regulation of GPR109A [32].

LPS was able to up-regulate GPR109A, and niacin treatment inhibited the LPS-induced up-regulation. Niacin was also found to prevent the nuclear translocation of p-NF-κB and subsequently inhibited the up-regulation of inflammatory cytokines in LPS-treated cells. Since the up-regulation of GPR109A occurs in PD, it is plausible that niacin may play an anti-inflammatory role in combating pro-inflammatory cytokine expression in PD by polarizing macrophages to the M2 phenotype [8]. An anti-inflammatory effect of niacin mediated by GPR109A has been shown in human monocytes as a model to correlate with antiatherogenic effects [33] but not in the context of PD.

Our study found that anti-inflammatory cytokines IL-1β and IL-6 were reduced with pre-treatment of niacin. Another anti-inflammatory cytokine, TNF-α did not respond to niacin treatment in vitro under these experimental conditions. Ordinarily, anti-inflammatory cytokines are a series of immunoregulatory molecules that control pro-inflammatory cytokines in a feedback loop. Some of these anti-inflammatory cytokines include IL-1 receptor antagonists, IL-4, IL-10, IL-11, and IL-13 [34]. However, the mechanism of this anti-inflammatory feedback loop fails under continuous insult by inflammatory stimuli leading to much higher levels of pro-inflammatory cytokines than levels of anti-inflammatory cytokines [35]. Specifically, IL-10, a potent anti-inflammatory cytokine, represses the expression of inflammatory cytokines such as TNF-α, IL-6 and IL-1β expressed by activated macrophages [36,37]. In addition, IL-10 can up-regulate other endogenous anti-inflammatory cytokines and down-regulate pro-inflammatory cytokines leading to phenotypic changes such as M1 macrophages to M2 Macrophages [8,37]. Therefore, IL-10 can act as a feedback regulatory mechanism and function in defense against pro-inflammatory cytokines at acute inflammation, but not in continuous onslaught of inflammatory cytokines, which occurs in chronic conditions of PD. Our studies of niacin intervention on PD subjects opens up new possibilities where niacin can be useful as an anti-inflammatory supplement in other pathological conditions where GPR109A is known to be up-regulated. The mechanism by which niacin exerts its effects is shown by determination of pro-inflammatory/anti-inflammatory profiles in PD patients before and after niacin therapy. In-vitro studies utilizing a murine macrophage cell line (RAW264.7) demonstrate the mechanism by which niacin reduces neuroinflammatory cytokines. Reduction of inflammatory cytokines in PD patients (either plasma or CSF) after niacin treatment would be more supportive.

To corroborate the mechanism of niacin action through the GPR109A receptor, we used specific siRNA against GPR109A to knock down the GPR109A expression substantially (~70%). A scrambled siRNA served as a negative control siRNA sequence, which did not affect niacin’s actions. With scrambled siRNA, niacin remained effective in blocking the LPS-induced inflammatory transcripts of IL-1β, and IL-6 but not TNF-α as described previously, similar to the effects observed in adipocytes by Digby et.al. [38]. It was not clear why TNF-α levels were not affected by niacin similarly to IL-1β and IL-6 levels. Our results revealed that niacin was unable to block the translocation of p-NF-κB in GPR109A knockdown cells, which indicates that blocking of p-NF-κB translocation to the nucleus is probably mediated through the GPR109A receptor. Additionally, GPR109A knockdown also blocked niacin’s ability to reduce the inflammatory cytokine expression. In summary, up-regulation of GPR109A denotes inflammation and niacin intervention is needed to suppress inflammation by preventing the p-NF-kB nuclear translocation signaling pathway. This evidence further supports that niacin inhibits the GPR109A up-regulation in PD patients, which may result in reduced neuroinflammation (Figure 7).

This study clearly indicates that GPR109A is essential for these anti-inflammatory actions of niacin. Knocking down GPR109A by the siRNA reduced niacin’s actions on p-NF-κB nuclear translocation and subsequent transcripts of inflammatory cytokines.

Neuroinflammation appears to be an important early potential target for disease modification in PD [39]. Inflammatory states can upregulate the protein expression of the receptor for niacin, GPR109A such as in PD patients where enhanced GPR109A expression is found in both the periphery [10] and in the substantia nigra [8]. Epidemiological study denotes that that risk of PD is lower in those consuming foods higher in niacin [40] and PD patients appear to have lower niacin levels and metabolites compared to controls shown here. Based on our findings and several other observations, we believe that niacin may be a better candidate to slow down PD progression by reducing neuroinflammation through GPR109A.

## 4. Materials and Methods

### 4.1. Antibodies and Reagents

Rabbit GPR109A antibody HM74 (T327) was purchased from Bioworld Technology Inc. Minneapolis, USA. Other reagents used in this study include β-actin antibodies (Sigma St. Louis, MO, USA), siRNA against GPR109A obtained from Qiagen (Qiagen, Germantown, MD, USA), BIO-RAD iScript cDNA synthesis Kit (Biorad, Carlsbad, CA, USA), and PCR Master Mix purchased from Thermo Scientific (Thermo Scientific, Waltham, MA, USA).

### 4.2. PD Subjects

Subjects participating in the study gave written informed consent under protocols approved by the Institutional Review Boards of Augusta University. Subjects were comprised of a control group and an age-matched PD group. Lab personnel accessed only de-identified data. Blood sample collection: Approximately 8 mL of whole blood was collected from the subjects in ethylene diamine tetraacetic acid (EDTA) tubes and kept on ice. Whole blood was spun at 2000 × *g* for 10 min in 15 mL tubes. Plasma was separated and stored at −80 °C. RBCs were collected and stored at −80 °C. An amount of 4 mL of ACK Lysing Buffer (Thermo Scientific, Waltham, MA, USA) was added to the WBC pellet to lyse any existing RBCs and spun again at 300 × *g* for 10 min. Supernatant was discarded from the clean WBC pellet. An amount of 1 mL of cold PBS was used to re-suspend the WBCs and spun at 3000 rpm for 5 min, then supernatant was discarded. Cell pellets were then stored at −80°C until further analyses.

### 4.3. HPLC Analyses

The quantitative analysis for niacin and its metabolites was performed at NMS Labs, Willow Grove, Pennsylvania using LC-MS/MS, Liquid Chromatography with tandem mass spectrometer detector. In summary, 25 µL of deuterated internal standard was added to a 0.20 mL aliquot of serum/plasma samples. Samples were pH-adjusted and extracted by solid phase extraction where samples were eluted with methanol, dried, and reconstituted with a formic acid in water solution, then transferred to vials for instrumental analysis. Analysis was performed on a Waters ACQUITY LC system with an Aquasil C18, 2.1 × 100 mm, 5.0 micron (part number 77505-102130), or equivalent USP L1 column, and TQ MS/MS detector. Quantitation was achieved by monitoring two transition ions following LC separation with positive-ion electrospray tandem mass spectrometry (LC-MS/MS) for each analyte and standard. Each analytical run was independently calibrated at concentrations of 10, 20, 50, 100, 400 and 500 ng/mL.

### 4.4. Cell Culture

BALB/c murine macrophage RAW264.7 cells were a gift from Dr. Krishnan Dhandapani, Augusta University. This murine macrophage cell line was chosen for this study because of its high expression of GPR109A and large availability compared to human macrophage cell lines which are SV40 T antigen modified; the genetic modification of human macrophages by SV40 T antigen may alter their morphology. Cells were cultured in RPMI medium 1640 (Life Technologies, Rockville, MD, USA), supplemented with 10% heat inactivated fetal bovine serum (FBS), 2 mM glutamine, and penicillin/streptomycin cocktail and incubated in a humidified incubator with 5% CO_2_ at 37 °C. Cells were treated with 400 μM niacin (Sigma-Aldrich, St. Louis, MO, USA) overnight followed by stimulation with LPS (20 ng/mL) for 20 min at 37 °C, 5% CO_2_. The expression of cytokine mRNA was determined by real time polymerase chain reactions (RT-PCR) and the expression of GPR109A, p-NF-κB, Histone 3 (H3), Rho-GDI alpha (ρGDi-α) and β-actin was confirmed using respective antibodies in Western blot assays.

### 4.5. siRNA Transfection

RAW264.7 cells were cultured in a six-well tissue culture dish for 24 h prior to transfection, at 70%–80% confluence. Scrambled siRNA (negative control) and gene-specific GPR109A siRNA were used along with transfection reagent Lipofectamine2000 (Invitrogen, Carlsbad, CA, USA) to form DNA-reagent complexes. These complexes were allowed to sit at room temperature for 10 min according to the manufacturer’s instruction and were added to the cells, drop-wise, in six-well plates. After 6 h incubation at 37 °C, 5% CO_2,_ the medium was replaced with fresh RPMI 1640 media supplemented with 10% heat-inactivated FBS for an additional 24 h. Transfected cells were treated with 400 μM niacin overnight prior to LPS (20 ng/mL) for 20 min at 37 °C, 5% CO_2_. Cells were harvested followed by nuclear and cytosolic fractionations and subjected to Western blot to determine the level of p-NF-κB translocated to the nucleus under GPR109A knockdown conditions. Silencing of GPR109A was determined by Western blot assay.

### 4.6. Nuclear and Cytosolic Fractions

The murine macrophage RAW264.7 cells were seeded at 1 × 10^6^ cells per well in a six-well plate for 24 h resulting in approximately 80% confluence. Cells were replaced with fresh media containing 10% FBS and penicillin/streptomycin antibiotics with niacin at 400 µM overnight followed by treatment with LPS at 20 ng/mL for 20 min at 37 °C, 5% CO_2_. All RAW264.7 cells were treated with niacin and LPS in subsequent experiments unless specifically mentioned. Cells were washed twice with cold PBS after the above treatment and subjected to nuclear and cytosolic fractionations with modification described by Antalis et al. [41]. Briefly, cells were harvested with 475 µl of cold hypotonic buffer (10 mM Tris pH 7.5, 10 mM NaCl, 3 mM MgCl_2_, 0.1 mM EGTA, 1 mM DTT, 1 mM PMSF, 1 µg/mL Leupeptine, 1 µg/mL Pepstatin A and 1 µg/mL Aprotinin) containing protease and phosphatase inhibitor followed by addition of 0.05% NP40, briefly vortexed and centrifuged at 14,000 rpm for 40 s at 4 °C [41]. Supernatant was saved and used as a cytosolic fraction. The pellets were washed twice with cold hypotonic buffer containing protease and phosphatase inhibitor without NP40 followed by centrifugation at 14,000 rpm for 40 s at 4 °C. Finally, nuclear pellets were dissolved in 50 µl RIPA buffer (Thermos Scientific, Waltham, MA, USA) containing protease and phosphatase inhibitor and incubated for 30 min on ice prior to use.

### 4.7. Western Blot

Equal amounts of nuclear fractions were loaded into 4%–15% BIORAD SDS–PAGE gels and ran at 160 volts followed by transfer of protein to PVDF membrane as described in the manufacturer’s instruction manual. Briefly, membranes were incubated with 5% non-fat dry milk (NFDM) for 1 h at room temperature. After blocking, membranes were incubated with appropriate primary antibody concentration (1:1000) overnight followed by secondary antibody (1:5000) incubation with intermittent washing with PBS containing 0.05% Tween 20 and subsequently developed using Denville Scientific Inc. ECL kit (Swedesboro, NJ, USA). The analyses of band intensity were performed using Image J software.

### 4.8. Reverse Transcriptase Polymerase Chain Reaction Analysis

Total RNA was extracted with TRIzol reagent (Gibco, Grand Island, NY, USA) either in niacin, LPS or in combination. Total RNA (1 μg) was subjected to reverse transcription into cDNA synthesis using reverse transcriptase enzyme in a total 20 μL volume with the instructions described in kit manuals (BIO-RAD, Carlsbad, CA, USA). β-actin mRNA was used as the housekeeping gene for internal control. For PCR, briefly, each reaction volume was 25 μL containing 2 μL cDNA template, 10 pmol of each primer (forward and reverse), appropriate amount of DNAse/RNAse free water and 12.5 μL of Thermo Scientific Dream Taq Green PCR master mix (Waltham, MA, USA). The PCR cycling condition was 30 s of denaturation at 95 °C, 30 s of annealing at 55 °C, and 30 s of elongation at 72 °C over 30 cycles for all reactions. The following Primers were used in this study:

TNF-α Forward: 5′- ATA GCT CCC AGA AAA GCA AGC-3′; TNF-α Reverse: 5′- CAC CCC GAA GTT CAG TAG ACA-3′; IL-6 Forward: 5′- TGG AGT CAC AGA AGG AGT GGC TAA G-3′; IL-6 Reverse: 5′- TCT GAC CAC AGT GAG GAA TGT CCA C-3′; IL-1β Forward: 5′-GCC TTG GGC CTC AAA GGA AAG AAT C-3′; IL-1β Reverse: 5′-GGA AGA CAC AGA TTC CAT GGT GAA G-3′; β-actin Forward: 5′-TCA CCC ACA CTG TGC CCA TCT ACG A-3′; β-actin Reverse: 5′-GGA TGC CAC AGG ATT CCA TAC CCA-3′.

### 4.9. Immunostaining

RAW264.7 cells were plated in six-well plates, treated with niacin and LPS as described above before washing with PBS, followed by fixation with 4% paraformaldehyde in 0.1 M phosphate buffer at pH 7.2. Cells were blocked in 4% bovine serum albumin (BSA) in PBS for 1 h at room temperature and permeabilized with 0.2% Triton X-100 for 15 min. The cells were incubated with anti-p-NF-κB antibody at 4 °C overnight. After three washes, the cells were incubated with secondary antibodies conjugated to Alexa flour 488, and vectashield was added to slides containing DAPI dye for labeling nuclei. The immunoreactivity product was visualized with the Zeiss fluorescence inverted microscope, and a CCD camera video system (Dage-MTI Inc., Michigan City, IN, USA.) was used to capture fluorescent images with different optical filters. The captured images were stored and processed using Photoshop software (Adobe Systems, San Jose, CA, USA). The band intensity of western blot, PCR products and fluorescent microscopic data were quantitated by ImageJ software [42]. The nuclear translocation of p-NF-κB fluorescence was quantitated by subtracting the cytosolic fraction from the both p-NF-κB and DAPI mask at region of Interest (ROI) using ImageJ software.

### 4.10. Statistical Analysis

Gel and Western blot images were quantitated by densitometric analysis using Image-J. Data in the figures and text were expressed as means ± SD. Significant differences between groups were determined by two-tailed Student’s t-test at *p* < 0.05. Data analysis was performed in Microsoft Excel^R^ (Microsoft corporation v2016).

## 5. Conclusions

Niacin, although shown to be anti-inflammatory, has never been shown to reduce inflammation in Parkinson’s disease (PD). Neuroinflammation is a major contributing factor to the pathology of PD. Currently, there is no cure for PD and common interventions focus on restoring dopamine levels in the brain and/or treating motor or non-motor symptoms. We have demonstrated an up-regulation of G-protein coupled receptor A (GPR109A), a niacin receptor, in macrophages of PD subjects, suggesting an immune-regulatory role for this receptor in the pathogenesis of PD. This in vitro work and patient data substantiate an anti-inflammatory action of niacin via GPR109A in reducing the production of inflammatory cytokines through regulating phospho-NF-kB nuclear translocation. Treating neuroinflammation may open new avenues to combat motor and non-motor symptoms of PD patients by enhancing their quality of life.

## Figures and Tables

**Figure 1 ijms-20-04559-f001:**
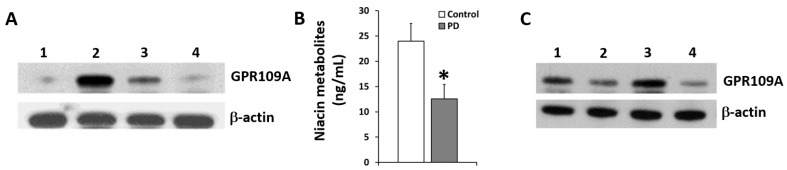
Expression of GPR109A protein in PBMC’s and neuronal cell line N27, and serum levels of niacin in healthy and PD patients. (**A**) Expression of GPR109A measured by immune-blot in white blood cells from control subject (lane 1), age matched PD subject (lane 2) and PD subject supplemented with niacin for half a month (lane 3) and one month (lane 4). β-actin is for loading control of total protein on SDS-PAGE gels. (**B**) Niacin metabolites in serum of PD subjects (*n* = 9), compared to the age-matched control subjects (*n* = 9). Data presented is Mean ± standard deviation (SD) (*n* = 18), * *p* < 0.05. (**C**) GPR109A expression (upper panel) in the N27 neuronal cell line. Untreated (lane 1), treated with 0.4 mM Niacin (lane 2), treated with MPP+ (lane 3) and after treatment with MPP+ (1 mM) in the presence of niacin (lane 4). β-actin (lower panel) was used as an internal loading control.

**Figure 2 ijms-20-04559-f002:**
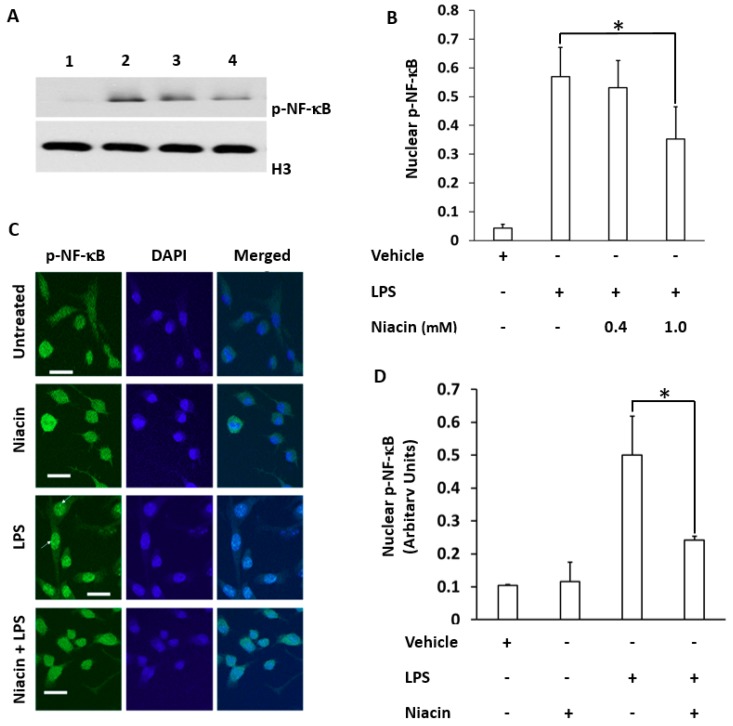
Inhibition of the translocation of LPS-induced phospho-Nuclear Factor-κB (p-NF-κB) into the nucleus of RAW264.7 cell line by niacin. (**A**) Western blot of p-NF-κB protein in nuclear fractions of RAW264.7 macrophages. No treatment (lane 1), treated with LPS (lane 2) and treated with LPS in the presence of 0.4 mM (lane 3) niacin and 1 mM niacin (lane 4). Histone-3 (H3) was used as a loading/nuclear marker. (**B**) Densitometric analysis of p-NF*-κB* protein normalized to H3 density. Data is presented as Mean ± SD levels of p-NF*-κB* from three independent experiments (*n* = 3), * *p* < 0.05. (**C**) Immuno-localization of punctate pattern of p-NF-κB in the nucleus of RAW264.7 cells before and after LPS stimulation, as well as in the presence and absence of niacin. White arrows show the nuclear translocation of p-NF-κB in RAW264.7 cells treated with LPS. Bar represents 100 μM. (**D**) p-NF-κB^+^ cell count normalized to total cells/ high powered field. Data is presented as Mean ± standard deviation (SD) from analysis of six separate high power cell images of at least 10 cells. * *p* < 0.05.

**Figure 3 ijms-20-04559-f003:**
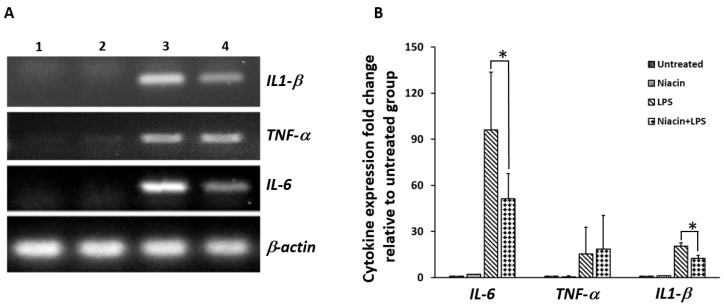
Effect of niacin on the expression of IL-6, TNF-α and IL-1β mRNA induced by LPS in RAW264.7 macrophage cell line by PCR analysis. (**A**) Expression of IL-6, TNF-α and IL-1β transcripts induced by LPS in the RAW264.7 macrophage cell line. Untreated cells (lane 1), treated with 0.4 mM niacin (lane 2), LPS (lane 3) nd LPS in the presence of niacin (lane 4). β-actin was used as an internal control. (**B**) Densitometric analysis showing the fold changes in expression of IL-6, TNF-α and IL-1β transcripts after normalization with β-actin. Data presented is Mean ± SD (*n* = 3). * *p* < 0.05.

**Figure 4 ijms-20-04559-f004:**
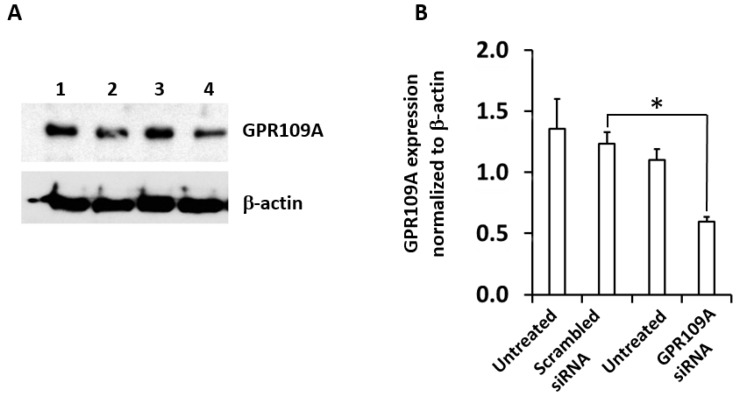
Knockdown of GPR109A protein expression in RAW264.7 cells after transfection with GPR109A-specific siRNA. (**A**) Western blot analysis to show the reduction in the expression of GPR109A protein (upper panel) in RAW264.7 cells, after transfection with *GPR109A*-specific siRNA. Untransfected cells (lanes 1 and 3), Scrambled siRNA (lane 2) and *GPR109A*-specific siRNA (lane 4), β-actin was used as a loading control (lower panel). (**B**) Densitometric analyses of Western blot band intensity of *GRP109A* protein levels were normalized to β-actin protein as an internal control. Data is presented as Mean ± SD of 3 replicates and * *p* < 0.05.

**Figure 5 ijms-20-04559-f005:**
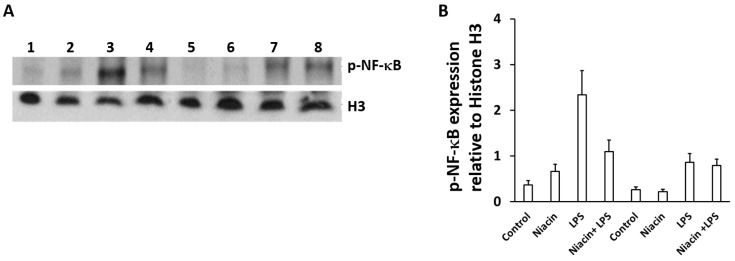
Knockdown of GPR109A inhibits the LPS-induced nuclear translocation of p-NF-κB protein in RAW264.7cells. (**A**) Western blot showing suppression of the nuclear translocation of p-NF-κB protein in macrophages transfected with scrambled siRNA (lanes 1–4) and GPR109A siRNAs (lanes 5–8), in untreated (lanes 1 and 5), treated with niacin (lanes 2 and 6) and LPS alone (lanes 3 and 7), and in combination with LPS and niacin (lanes (4 and 8). Histone 3 (H3) was used as a nuclear marker (A, lower panel). (**B**) GPR109A expression is presented as densitometric analysis and normalization with H3 band intensity. Data is presented as Mean ± SD of 3 replicates. * *p* < 0.05.

**Figure 6 ijms-20-04559-f006:**
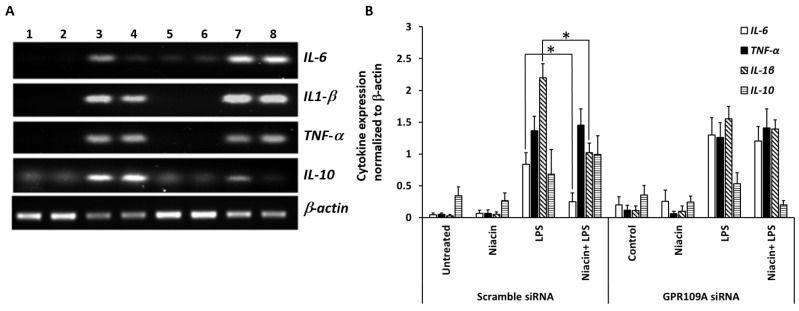
Expression of pro- and anti-inflammatory cytokine transcripts in RAW264.7 cells after GPR109A knockdown and LPS stimulation in the presence and absence of niacin. (**A**) Gene expression analysis of IL-1β, IL-6, TNF-α, and IL-10 transcripts in RAW264.7 cells transfected with scrambled (lanes 1–4) or GPR109A siRNA (lanes 5–8). Treatments groups were untreated controls (lanes 1, 5), niacin only (lanes (2, 6), LPS only (lanes 3, 7) and LPS in the presence of niacin (lanes 4, 8). (**B**) Densitometric analysis of changes in the expression levels of cytokine genes relative to β-actin. Data presented as Mean ± SD of 3 replicates. * *p* < 0.05.

**Figure 7 ijms-20-04559-f007:**
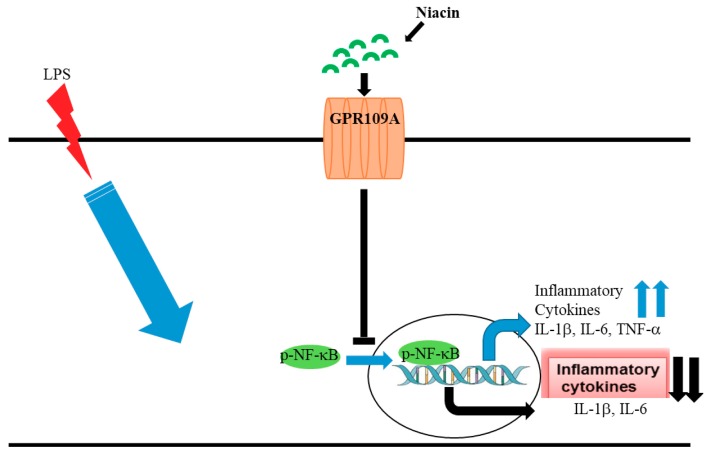
Inhibitory activity of niacin on LPS induced p-NF-κB signaling cascade and transcriptional regulation of inflammatory cytokines.

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
