# Peer review of "Niacin Ameliorates Neuro-Inflammation in Parkinson’s Disease via GPR109A"

_ijms, 2019, doi:10.3390/ijms20184559_

Round 1
Reviewer 1 Report
Manuscript ID: ijms-584610
Title: Niacin Inhibits inflammation Through GPR109A in Murine Macrophages
General comment
The manuscript from Giri et al. describes the anti-inflammatory properties of niacin in murine macrophages and give an insight on the underlying mechanism.
The manuscript is well written, the experiments have been well conducted, and the results are clearly presented.
The manuscript could be of interest and is worth of consideration to be published, after major revision listed below.
Major
1) Why the cytokine profile has been studied at mRNA level only? Authors should evaluate the secretion of cytokines.
2) Niacin has been used at a pretty high concentration (400 µM). Authors should give an explanation of this choice, and a comparison with literature data. Moreover, did the authors check for potential cytotoxic effects?
Minor
Page 3 line 123: authors should use “concentration” and avoid the use of “dose” when describing in vitro experiments
Page 7 line 224: please correct “ant-inflammatory”
Page 8 lines 259-263: please be careful that these lines are part of the caption of Figure 7
Page 10 line 349 (and throughout the manuscript): “ml” should be replaced by “mL”
Author Response
We thank the reviewers for their thoughtful, thorough, and constructive comments. We have extensively modified the manuscript and have addressed all the concerns. As suggested, the revised portion shows track changes. Please see the detailed response below after specific comments.
Reviewer 1:
General comment
The manuscript from Giri et al. describes the anti-inflammatory properties of niacin in murine macrophages and give an insight on the underlying mechanism.
The manuscript is well written, the experiments have been well conducted, and the results are clearly presented.
The manuscript could be of interest and is worth of consideration to be published, after major revision listed below.
Response: We thank the reviewer for the encouraging comments.
1) Why the cytokine profile has been studied at mRNA level only? Authors should evaluate the secretion of cytokines.
Response: We are in full agreement with the comments. The reason we conducted mRNA levels is solely because a wide variety of studies show strong correlation between mRNA expression and secreted protein levels. mRNA expression signatures are mostly used to surrogate measure of its translation products for cellular function and pathway changes are time dependent. The aim of our studies is to evaluate the effect of niacin in response to lipopolysaccharide (LPS) treatment in a short time period of 20min, before the toxic effect of LPS could occur. Therefore, authors thought measuring expression of mRNA level was more appropriate in the context of the experimental design. We will certainly incorporate this suggestion in our follow up studies.
2) Niacin has been used at a pretty high concentration (400 µM). Authors should give an explanation of this choice, and a comparison with literature data. Moreover, did the authors check for potential cytotoxic effects?
Response: We agree that micro molar concentrations in the in-vitro studies do not appear physiological. However, our preliminary experiments on niacin dosage optimization studies suggested that niacin at 400 µM optimally blocked phospho-NF-κB translocation to the nucleus in the Raw264.7 cell line. Although we did not measure cellular toxicity, we observed no apparent toxicity throughout the experiments. This was in concurrence with the published literature cited below that used niacin at much higher concentration up to 3 mM and reported no cellular toxicity.
References:
Huang H, Koelle P, Fendler M, Schroettle A, Czihal M, et al. (2014) Niacin Reverses Migratory Macrophage Foam Cell Arrest Mediated by oxLDL In Vitro. PLoS ONE 9(12): e114643. doi:10.1371/journal.pone.0114643Yanhong Si, Ying Zhang, Jilong Zhao, Shoudong Guo, Lei Zhai, Shutong Yao, Hui Sang, Nana Yang, Guohua Song, Jue Gu, and Shucun Qin. Niacin Inhibits Vascular Inflammation via Downregulating Nuclear Transcription Factor-κB Signaling Pathway. Mediators of Inflammation Volume 2014, Article ID 263786
http://dx.doi.org/10.1155/2014/263786
Kim SW, Lee JH, Moon JH, Nazim UM, Lee YJ, Seol JW, Hur J, Eo SK, Lee JH, Park SY. Niacin alleviates TRAIL-mediated colon cancer cell death via autophagy flux activation. Oncotarget. 2016 Jan 26; 7(4):4356-68.
Page 3 line 123: authors should use “concentration” and avoid the use of “dose” when describing in vitro experiments
Response: We have made the correction.
Page 7 line 224: please correct “ant-inflammatory”
Response: We have made the correction.
Page 8 lines 259-263: please be careful that these lines are part of the caption of Figure 7
Response: Revised.
Page 10 line 349 (and throughout the manuscript): “ml” should be replaced by “mL”
Response: Revised.
Reviewer 2 Report
While this is an interesting research topic, I feel that the manuscript needs several improvements before it can be accepted.
-The title doesn't represent the information presented in the manuscript, because you have PD human subjects if I understood well. So, consider changing the title.
-Regarding the abstract, I consider it is well written and it represents the basic information of your manuscript.
-Regarding the introduction, although the substantial part it is well written there are several paragraphs that have to add in the discussion. Furthermore, consider revising acronym like Alzheimer's Disease line 60.
-Regarding the results, have to describe only the results and avoid to add information that is for the discussion. Mainly, the first sentence of each result section. Furthermore, you have to improve the image and the image title of each figure result since they are of very low quality, the histograms as well. Consider eliminating the references of the figure legends and add in another section. Furthermore, consider revising the concepts of protein levels and mRNA expression because you made a mistake with these two, in fact, you put in the other way around. Likewise, the gene names are wrong, you have to put in italic...
-Figure 7 is not a result, just a summary figure so you have to add at the end of the discussion. Besides, consider improving the quality of this image.
-Regarding the discussion, in my opinion, is the poor part of the manuscript, consider comparing your results with other studies, it seems that you are the only group that works with niacin. Likewise, consider revising english grammar, acronyms and several concepts. Besides, is too much speculative, especially, from line 317 to 325, consider eliminating or changing this paragraph.
-Interesting research that after all these changes the quality of the manuscript will be improved.
Thank you so much
Author Response
We thank the reviewers for their thoughtful, thorough, and constructive comments. We have extensively modified the manuscript and have addressed all the concerns. As suggested, the revised portion shows track changes. Please see the detailed response below after specific comments.
Reviewer 2:
-The title doesn't represent the information presented in the manuscript, because you have PD human subjects if I understood well. So, consider changing the title.
Response: This is an excellent suggestion and we have appropriately changed the title of the manuscript to Niacin Ameliorates Neuro-inflammation in Parkinson’s Disease via GPR109A.
-Regarding the introduction, although the substantial part it is well written there are several paragraphs that have to add in the discussion. Furthermore, consider revising acronym like Alzheimer's Disease line 60.
Response: We thank the reviewer for this comment. We have moved certain portions from introduction to the discussion. We have now added the acronym for Alzheimer's Disease. We strongly feel that introduction reads much better. We appreciate the comment.
-Regarding the results, have to describe only the results and avoid to add information that is for the discussion. Mainly, the first sentence of each result section. Furthermore, you have to improve the image and the image title of each figure result since they are of very low quality, the histograms as well. Consider eliminating the references of the figure legends and add in another section. Furthermore, consider revising the concepts of protein levels and mRNA expression because you made a mistake with these two, in fact, you put in the other way around. Likewise, the gene names are wrong, you have to put in italic...
Response: We agree with all these comments and have made changes accordingly.
We have now revised the figures completely and improved the quality and resolution fit for publication. The reference has been removed from the legends. We have extensively revised the figure legends to clearly differentiating the protein levels from mRNA levels. The gene names are not italicized in the appropriate figures.
-Figure 7 is not a result, just a summary figure so you have to add at the end of the discussion. Besides, consider improving the quality of this image.
Response: Figure 7 is moved at the end of the discussion and the quality of the image is improved. We thank reviewer for this comment.
-Regarding the discussion, in my opinion, is the poor part of the manuscript, consider comparing your results with other studies, it seems that you are the only group that works with niacin. Likewise, consider revising English grammar, acronyms and several concepts. Besides, is too much speculative, especially, from line 317 to 325, consider eliminating or changing this paragraph.
Response: We have made substantial changes to discussion accordingly. We have now removed the speculative sentences from the area. We hope that the reviewer finds the revised section appropriate.
Round 2
Reviewer 1 Report
The authors replied to all the concerns raised and improved the manuscript.
Therefore, I suggest to publish the revised version as is.